# Analysis of Characteristics and Suppression Methods for Self-Defense Smart Noise Jamming

**Yongzhe Zhu, Zhaojian Zhang, Binbin Li \*, Bilei Zhou, Hao Chen and Yongliang Wang**

Early Warning Academy, Huangpu Road, Wuhan 430019, China; jackychu27@163.com (Y.Z.); zzj554038@163.com (Z.Z.); zhoubilei666888@sina.com (B.Z.); ch19930922@163.com (H.C.); ylwangkjld@163.com (Y.W.)
**\*** Correspondence: binbinli_1025@163.com

**Abstract:** Self-defense smart noise jamming can automatically aim a signal frequency and obtain antenna gain and matching filtering processing gain, posing a huge threat to the normal performance of radar. In response to this situation, this article conducts an in-depth analysis of two typical smart noise jamming methods: noise convolution jamming methods and noise product jamming methods. The distribution characteristics of the two jamming methods in both time–frequency dimensions and their internal time–frequency relationships are analyzed. Based on this, a self-defense smart noise jamming suppression method based on pulse frequency stepping is proposed. This method obtains the true distance information of the target based on the phase difference caused by frequency stepping between adjacent pulses and uses this information to construct a filter to filter the radar echo, achieving jamming suppression. Simulation experiments have verified the effectiveness of this method.

**Keywords:** pulse frequency stepping; self-defense smart noise jamming; analysis of jamming characteristics; jamming suppression





## 1. Introduction

As it plays an important role in battlefield information acquisition and situational awareness, radar has always been faced with the threat of electronic jamming [1–3]. In recent years, with the development of digital radio frequency memory(DRFM) [4,5] technology, a jammer can rapidly intercept radar signals, modulate them, and forward them. For this reason, many new jamming styles have emerged, including shift-frequency jamming [6,7], interrupted sampling and repeater jamming (ISRJ) [8–10], and smart noise jamming [11–16]. The above-mentioned jamming methods are all coherent with radar-transmitted signals and can obtain gain from matched filtering and pulse compression processing [17]. When jammings enters from the main lobe, it can also gain antenna main lobe gain, with energy much stronger than the real target signal, greatly increasing the difficulty of accurately detecting targets using radar. Among these methods, shift-frequency jamming and ISRJ mainly have deceptive effects, and the resultant processing difficulty is relatively small. Recently, scholars have proposed a variety of suppression methods, but smart noise jamming can make full use of the transmitted signal power of the radar, and it can have a simultaneous suppressive effect on the signal in the time domain and the frequency domain, which is more difficult to suppress. In Reference [18], a smart noise convolution jamming suppression method based on the FDA-MIMO radar system was proposed. A range–angle two-dimensional matched filter was used to suppress the range of the mismatched main lobe jamming. However, this method requires a high signal-to-jamming-noise ratio (SJNR) of the target and a priori information on the target's motion state. The actual situation may not be so ideal. In Reference [19], a smart noise jamming suppression method based on optimized atomic dictionary decomposition was proposed. This method obtains jamming signal information via dual-channel waveforms and then decomposes the jamming parameters by using atomic dictionary decomposition to suppress

it. However, this method can only handle jamming with lower jamming-to-signal ratios (JSR) and cannot accurately obtain jamming parameters when the jamming JSR is high and suppressive. In Reference [20], a smart noise suppression method for fast fractional filtering was proposed. This method performs a fractional Fourier transform (FrFT) on the signal, selects the optimal fractional transform rotation angle, extracts target parameters based on the fractional fourth order origin moment, and designs a filter in the fractional domain to filter out jamming. Finally, an inverse FrFT transform is performed to restore the target signal; however, for strong suppressing jamming, it is difficult to distinguish jammings from target signals in the fractional domain. In Reference [21], a smart noise jamming model of airborne phased array was modeled, and corresponding suppression methods were proposed; however, this method can only suppress sidelobe jamming. The authors of Reference [22] designed a waveform with a "pushpin" shape for an ambiguity function based on orthogonal diversity technology to suppress smart noise jamming. However, this method requires the signal intercepted by the DRFM jammer to lag by one pulse repetition period relative to the radar transmission signal. In summary, the above methods will be difficult to apply to self-defense smart noise jamming that is intercepted, modulated, and forwarded within a pulse repetition cycle.

Therefore, based on the properties of convolution and product operations in the Fourier transform, this paper analyzes the correlation between smart noise jamming and target signals. It is found that noise product jamming and noise convolution jamming are actually modulated by a series of continuous, random-amplitude impulse functions in the frequency and time domains, respectively, and their inherent time–frequency relationships remain unchanged. Based on the above conclusion, this article proposes a self-defense smart noise jamming suppression method based on pulse frequency stepping. This method utilizes the phase difference caused by frequency stepping between pulses and obtains the true distance information of the target in the pulse domain. Using this information, a narrowband filter is constructed to filter the mixed signal of the target and the jamming, achieving jamming suppression.

## 2. Analysis of the Characteristics of Smart Noise Jamming

Assuming that the target itself carries a jammer which intercepts the radar signal, the jammer uses pre-designed noise templates [12] $n_c(t)$ and $n_p(t)$ for convolution or noise modulation. The noise convolution jamming can be represented as follows:

$$J_c(t) = s(t) * n_c(t) \tag{1}$$

where $*$ represents the convolution operation. When the noise time width is $\tau$, noise can be regarded as the sum of all impulse signals of random amplitude over its duration, namely:

$$n_c(t) = \sum_{i=0}^{\tau} A_i \cdot \delta(t - i) \tag{2}$$

where $\delta(t)$ represents the impulse signal, $A_i$ represents the amplitude of the $i$-th impulse signal, and Equation (1) can be expressed as follows:

$$J_c(t) = s(t) * n_c(t) = \sum_{i=0}^{\tau} A_i \cdot \delta(t - i) * s(t) = \sum_{i=0}^{\tau} A_i \cdot s(t - i) \tag{3}$$

According to the property of the impulse function, the essence of noise convolution jamming in the time domain is to carry out the delay modulation and amplitude modulation of radar signal with a series of continuous impulse functions of random amplitude within the duration of the noise. The linear frequency-modulated (LFM) signal commonly used in radar is a singer function similar to a "point" after pulse compression, so the noise convolution jamming after pulse compression is distributed behind the signal, and the duration depends on the noise duration.

The noise product jamming can be expressed as follows:

$$J_p(t) = s(t) \cdot n_p(t) \tag{4}$$

From Equation (4), it can be seen that noise product jamming is a random amplitude modulation of the signal in the time domain.

Transforming two types of jamming signals into the frequency domain. Assuming that the spectra of $s(t)$, $n_c(t)$, $n_p(t)$, $J_c(t)$, and $J_p(t)$ are $S(f)$, $N_c(f)$, $N_p(f)$, $J_c(f)$, and $J_p(f)$, respectively. According to the properties of the Fourier transform, the spectrum of noise convolutional jamming is as follows:

$$J_c(f) = S(f) \cdot N_c(f) \tag{5}$$

From Equation (5), it can be seen that noise convolutional jamming is a random amplitude modulation of the radar signal spectrum in the frequency domain.

The spectrum of noise product jamming is as follows:

$$J_c(f) = S(f) * N_p(f) \tag{6}$$

When the noise bandwidth is $B$, the noise spectrum can be regarded as the sum of all impulse signals of random amplitude within the bandwidth:

$$N_p(f) = \sum_{i=-B/2}^{B/2} A_i \cdot \delta(f - i) \tag{7}$$

In this case, Equation (6) can be expressed as follows:

$$J_c(f) = S(f) * N_p(f) = S(f) * \sum_{i=-B/2}^{B/2} A_i \cdot \delta(f - i) = \sum_{i=-B/2}^{B/2} A_i \cdot S(f - i) \tag{8}$$

As can be seen from Equation (8), the essence of noise product jamming in the frequency domain is to carry out spectrum shift and amplitude modulation of the radar signal with a series of continuous and random amplitude impulse functions within the noise bandwidth. Due to the range–Doppler effect in the LFM signal, a spectrum shift will cause a time domain delay, so the noise product jamming after pulse pressure is distributed before and after the signal in the time domain, and the duration depends on the noise signal bandwidth.

It is worth mentioning that the jamming scenario assumed in this paper is that the jammer stores, modulates, and forwards the intercepted radar signal in a pulse repetition period to achieve the suppressive jamming effect. Considering the actual engineering implementation requirements, it is necessary to make noise templates in advance [12] in order to quickly modulate signals. Without a loss of generality, consider the radar transmitting signal an LFM signal, a signal time width of 1 ms, and a bandwidth of 5 MHz. In order to better analyze the characteristics of smart noise jamming, for noise convolution jamming, the LFM signal is convolutional modulation by Gaussian white noise with a duration of 300 μs, and for noise product jamming, the LFM signal is product modulated by Gaussian noise with a bandwidth of 0.6 MHz, and the JSR is 0 dB.

Figures 1 and 2 show a time–domain comparison diagram of the waveform and spectrum, respectively, after the pulse compression of the two jamming signals and the intercepted radar signals. For the convenience of observation, the signals are set at the 3000th distance gate (all experiments here and in this paper were simulated in MATLAB).

The simulation results show that in the time domain, the noise convolution jamming energy is very strong, but the jamming will be delayed after the target signal. The noise product jamming signal energy is relatively weak, but it covers the whole target signal, and it is distributed before and after the target signal. In the frequency domain, both kinds of jamming can cover the whole target signal, and the noise convolution jamming has strong energy. In general, these two kinds of jamming can achieve strong suppression of the signal when the JSR is high; whether it is in the time domain or frequency domain, it is difficult for the traditional filtering method to play an effective role.

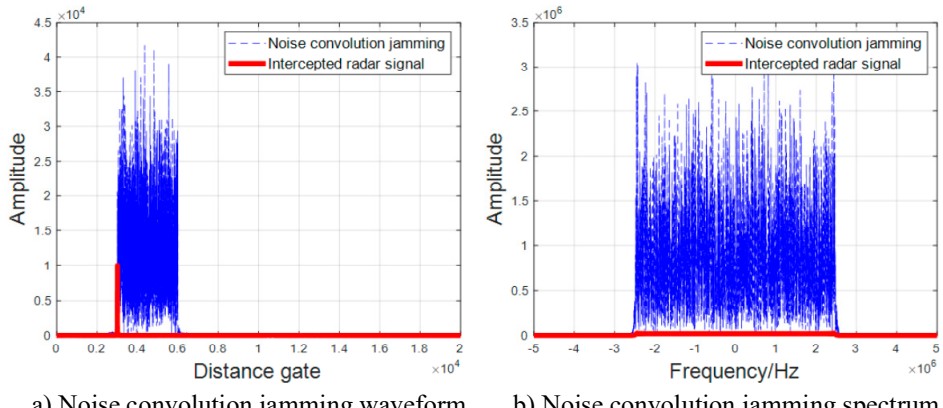

a) Noise convolution jamming waveform          b) Noise convolution jamming spectrum

**Figure 1.** Waveform and spectrum of noise convolutional jamming.

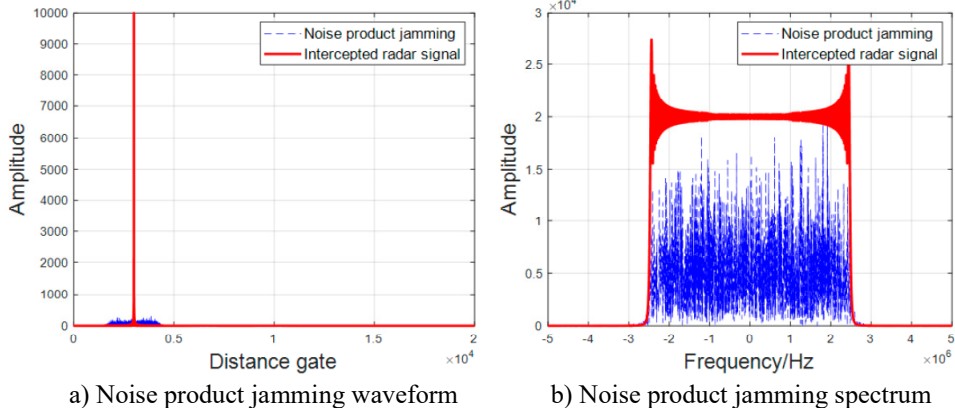

a) Noise product jamming waveform          b) Noise product jamming spectrum

**Figure 2.** Waveform and spectrum of noise product jamming.

### 3. Frequency Stepping Signal Model and Feasibility Analysis of Jamming Suppression

According to the analysis in the above section, strong smart noise jamming can cover signals in both the time domain and frequency domain, and it is difficult to suppress jamming in traditional single-time-domain processing or combined time–frequency-domain processing, so it is necessary to explore new processing domains. Adding a tiny step frequency between adjacent pulse carrier frequencies can cause a phase difference to exist between adjacent pulses which reflects the distance information of the target [23]. In order to distinguish it from the time domain distance, this distance information is called spatial distance in this paper. This provides a new way to suppress smart noise jamming. If the real distance information of a target can still be obtained from the phase difference between pulses in the disturbed state, the filter can be constructed according to the information so as to suppress smart noise jamming. The authors of Reference [23] proved the effectiveness of this method for forwarding range deception jamming, and determining whether it is effective for smart noise jamming requires further exploration.

A time–frequency analysis is a powerful tool for analyzing non-stationary signals which can intuitively show the law of the frequency components of signals changing with time [24]. The short time Fourier transform (STFT) is a commonly used linear time–frequency transformation method which is simple to calculate and does not generate cross terms. This paper uses the STFT method and a LFM signal as a reference to deeply analyze the time–frequency characteristics of smart noise jamming. The time–frequency distribution of $s(t)$ can be expressed as follows:

$$\Omega_s(t, f) = \Gamma\{s(t)\} = \int_{-\infty}^{\infty} s(\tau)\omega(\tau - t)e^{-j2\pi f\tau}d\tau \tag{9}$$

where $\Gamma\{\cdot\}$ represents the STFT operation, and $\omega(\cdot)$ represents the window-function in the STFT. Similarly, the time–frequency distribution of noise convolutional jamming and noise product jamming can be obtained as follows:

$$\Omega_{jc}(t,f) = \Gamma\{J_c(t)\} = \int_{-\infty}^{\infty} J_c(\tau)\omega(\tau - t)e^{-j2\pi f\tau}d\tau \tag{10}$$

$$\Omega_{jp}(t,f) = \Gamma\{J_p(t)\} = \int_{-\infty}^{\infty} J_p(\tau)\omega(\tau - t)e^{-j2\pi f\tau}d\tau \tag{11}$$

Figures 3–5 show the time–frequency distribution of the LFM signal, noise convolutional jamming, and noise product jamming, respectively.

As shown in the time–frequency distribution diagram, if the LFM signal is taken as a sample for reference, it can be found that the noise convolution jamming is obtained via the "backward" translation of the LFM signal following amplitude modulation in the time domain, and the noise product jamming is obtained via an "up and down" translation following amplitude modulation in the frequency domain, but the time–frequency relationship within the jamming itself does not change relative to the LFM signal. In order to better observe the time–frequency characteristics of smart noise jamming for different carrier frequencies, it is assumed that three LFM signals with large carrier frequency differences are transmitted at the same time. In this case, the time–frequency distribution diagram of signal and jamming is as follows.

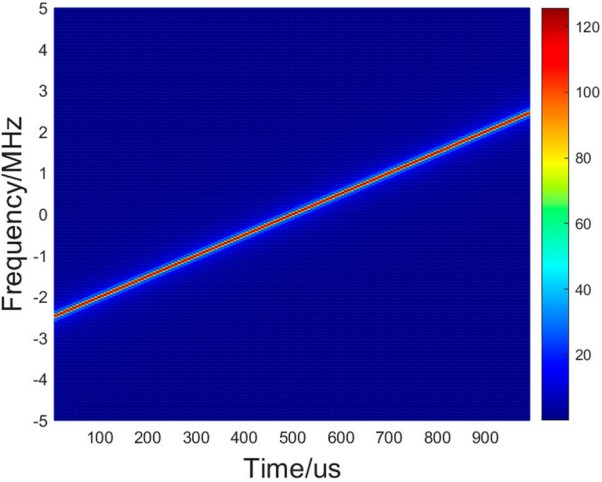

**Figure 3.** Time–frequency distribution diagram of LFM signal.

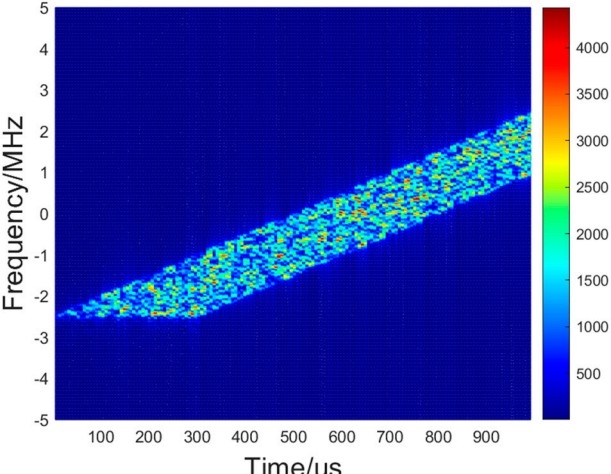

**Figure 4.** Time–frequency distribution diagram of noise convolutional jamming.

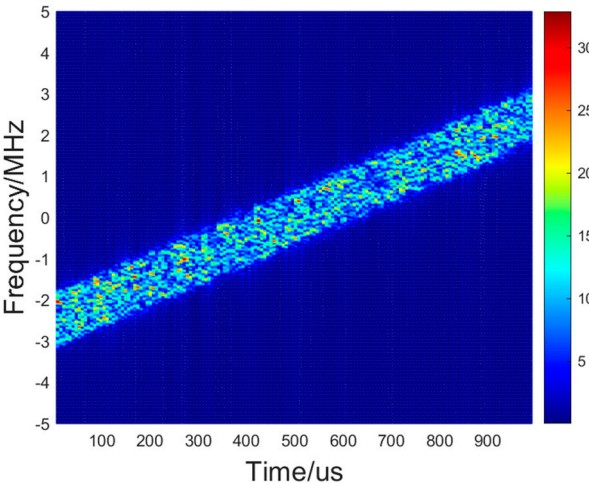

**Figure 5.** Time–frequency distribution diagram of noise product jamming.

It can be seen from the time–frequency distribution diagram (Figures 6–8) of multi-frequency noise convolution jamming and multi-frequency noise product jamming that the jamming has similar time–frequency characteristics to the signal, if the spatial distance can be reflected in the phase relationship of the multi-frequency signal, it can also be reflected in the jamming signal.

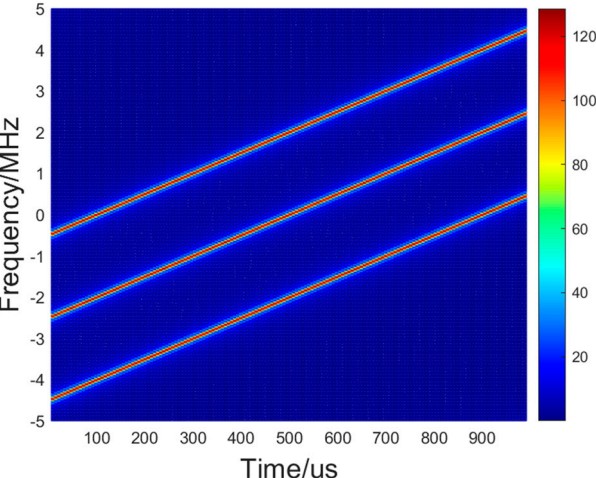

**Figure 6.** Time–frequency distribution diagram of multi-frequency LFM signal.

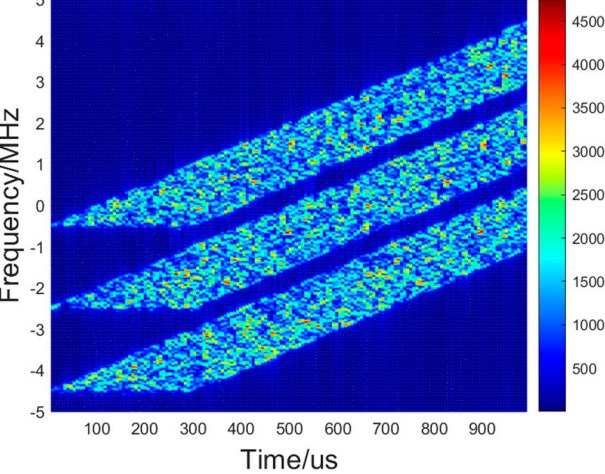

**Figure 7.** Time–frequency distribution diagram of multi-frequency noise convolutional jamming.

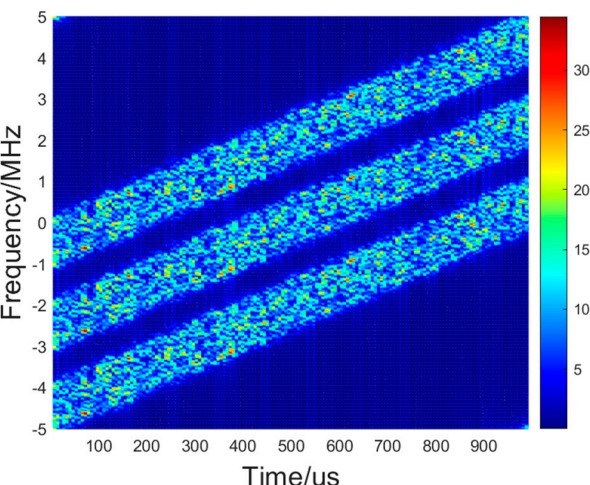

**Figure 8.** Time–frequency distribution diagram of multi-frequency noise product jamming.

Therefore, consider the time-sharing transmission of multi-frequency signals, that is, the transmission of signals with different carrier frequencies between pulses. The transmitted signal is as follows:

$$s(t) = \sum_{m}^{M} A_m \text{rect}(\frac{t}{\tau}) \exp(\text{j}\pi k t^2) \exp(\text{j}2\pi f_m t) \tag{12}$$

where $A_m$ represents the amplitude of the transmitting signal of the $m$-th pulse, $f_m$ is the carrier frequency of the $m$-th pulse, and $\text{rect}(\frac{t}{\tau}) = \begin{cases} 1, 0 < t \leq \tau \\ 0, else \end{cases}$.

Assuming uniform carrier frequency stepping between pulses, the carrier frequency of the transmitted signal of the $m$-th pulse is as follows:

$$f_m = f_0 + (m+1)\Delta f \tag{13}$$

where $\Delta f$ represents the step frequency.

Assuming there is a target at a distance $R$, the echo of the $m$-th pulse after down conversion processing is as follows:

$$s_r^m(t) = A'_m \text{rect}(\frac{t - 2R/c}{\tau}) \exp(-\text{j}2\pi f_m \frac{2R}{c}) \exp[\text{j}\pi k(t - \frac{2R}{c})^2] \tag{14}$$

where $A'_m$ represents the amplitude of the echo signal.

Similarly, it can be concluded that the echo of the $(m-1)$-th pulse signal after down conversion processing is as follows:

$$s_r^{m-1}(t) = A'_{m-1} \text{rect}(\frac{t - 2R/c}{\tau}) \exp(-\text{j}2\pi f_{m-1} \frac{2R}{c}) \exp[\text{j}\pi k(t - \frac{2R}{c})^2] \tag{15}$$

Thus, it is easy to obtain:

$$s_r^m(t) = s_r^{m-1}(t) \exp(-\text{j}2\pi \frac{2R}{c} \Delta f) \tag{16}$$

From Equation (16), it can be seen that the target signal has a fixed phase difference between adjacent pulses $-2\pi\frac{2R}{c}\Delta f$; this phase difference reflects the spatial distance $R$ of the target. According to the above analysis, the jamming is essentially the amplitude modulation and shift of the target signal in the time domain and frequency domain, and its internal time–frequency relationship does not change, so the jamming has the same phase difference between adjacent pulses, and the phase difference also reflects the spatial distance $R$ of the target.

In summary, there are two conclusions regarding smart noise jamming:

1.  Both noise convolution jamming and noise product jamming can cover the target signal in the time domain; the noise convolution jamming method is distributed behind the target signal, and the duration is determined by the noise duration. The noise product jamming is distributed before and after the target, and the duration is determined by the noise bandwidth.
2.  Noise convolution jamming and noise product jamming are essentially generated via the superposition of the amplitude modulation and translation of signals in the time and frequency domains. Noise modulation does not damage the time–frequency characteristics and phase relationship of the signal itself.

## 4. Main Lobe Smart Noise Jamming Suppression Method

According to the analysis in the above section, the most important part of this algorithm is the means of obtaining the spatial distance of the target. In fact, increasing the step frequency between pulses is equivalent to sampling in the pulse carrier frequency domain: the sampling interval is the step frequency, and the number of pulses is the number of sampling. When the pulse carrier frequency is uniformly stepped, the phase difference between the pulses is fixed, and the signal is equivalent to a single-frequency point signal in pulse dimension:

$$s(m) = \exp[-j2\pi \frac{2R}{c}(m-1)\Delta f] \tag{17}$$

The frequency of the signal is $\frac{2R}{c}$. To obtain the frequency, a simple method is to complete a fast Fourier transform (FFT) of the multiple pulses according to the distance gate; however, the accuracy of the FFT is limited by the number of sampling points and the sampling interval. Assuming that $M$ pulses are emitted, the accuracy of the FFT is $\frac{1}{M\Delta f}$, and the frequency is substituted into the precision as follows:

$$\frac{2R}{c} = \frac{1}{M\Delta f} \Rightarrow R = \frac{c}{2M\Delta f} \tag{18}$$

As can be seen from Equation (18), the spatial distance detection accuracy obtained via this method is $\frac{c}{2M\Delta f}$. Assuming that 16 pulses are emitted and the step frequency between pulses is 1 kHz, the spatial distance detection accuracy at this time is 9.375 km. This distance detection accuracy is very low. To improve the accuracy, either increase the step frequency or increase the number of pulses. However, a stepping frequency that is too large will produce relatively serious distance ambiguity, the principle of which is discussed in detail in Reference [23] and will not be repeated here. In practice, the number of pulses emitted by radar is also limited. Therefore, using the FFT method to obtain the spatial distance of the target is limited. In this paper, the concept of spectral estimation is used to estimate the spatial distance of the target.

Since there is a fixed phase difference between pulses that reflects the spatial distance, it can be considered that there is a steering vector about the spatial distance between pulses. Suppose there is a target at a distance $R_0$, and the steering vector of the target is as follows:

$$\boldsymbol{a}_t(R_0) = [1; e^{-j2\pi \frac{2R_0}{c}\Delta f}; e^{-j2\pi \frac{2R_0}{c}2\Delta f}; \dots; e^{-j2\pi \frac{2R_0}{c}(m-1)\Delta f}] \tag{19}$$

According to the analysis in Section 3, smart noise jamming is essentially the amplitude modulation and shift of the intercepted radar signal in the time domain and frequency domain, and its internal time–frequency relationship does not change; therefore, the jamming has the same steering vector as the target:

$$\boldsymbol{a}_J(R_0) = [1; e^{-j2\pi \frac{2R_0}{c}\Delta f}; e^{-j2\pi \frac{2R_0}{c}2\Delta f}; \dots; e^{-j2\pi \frac{2R_0}{c}(m-1)\Delta f}] \tag{20}$$

Assuming that the radar emits *M* pulses, the echo received by the radar can be expressed as follows:

$$X(t) = \sum_{m=1}^{M} a_t(R_0) \cdot s_r^m(t) + a_J(R_0) \cdot J^m(t) + N(t) \tag{21}$$

where $J^m(t)$ represents the jamming signal received at the *m*-th pulse repetition period. In order to facilitate representation, the noise convolution jamming and noise product jamming are uniformly represented as $J^m(t)$, and $N(t)$ represents the Gaussian white noise.

The covariance matrix of the echo data can be calculated as follows:

$$\hat{R}_x = \sum_{i=1}^{L} X(t) \cdot X(t)^H \tag{22}$$

where *L* represents the shotsnaps and $(\cdot)^H$ represents conjugate transpose operation.

By performing eigenvalue decomposition on $\hat{R}_x$, we have:

$$\hat{R}_x = U_S \Sigma_S U_S^H + U_N \Sigma_N U_N^H \tag{23}$$

where $U_S$ is the signal subspace corresponding to large eigenvalues, and $U_N$ is a noise subspace corresponding to small eigenvalues.

This paper adopts the multiple signal classification (MUSIC) algorithm [25] to conduct a one-dimensional search for the spatial distance of the signals:

$$P_{MUSIC}(R) = \frac{1}{a^H(R)U_N U_N^H a(R)} \tag{24}$$

where *R* is the spatial distance scanning range.

After obtaining the target spatial distance information via the MUSIC algorithm, a time domain filter can be set based on this information, and the mixed signal after pulse compression can be filtered to suppress jamming.

Figure 9 shows the processing flow of this paper to suppress the main lobe smart noise jamming. First, the received radar echo is divided into two channels: one is matched and filtered as the echo data to be filtered, and one is used to construct the covariance matrix. Secondly, the noise subspace is extracted from the covariance matrix eigendecomposition of a matrix. Then, the MUSIC algorithm is used to search the space distance to obtain the target space distance. Finally, a time domain filter is set after obtaining the space distance and filtering the echo data from another matched filter to achieve jamming suppression.

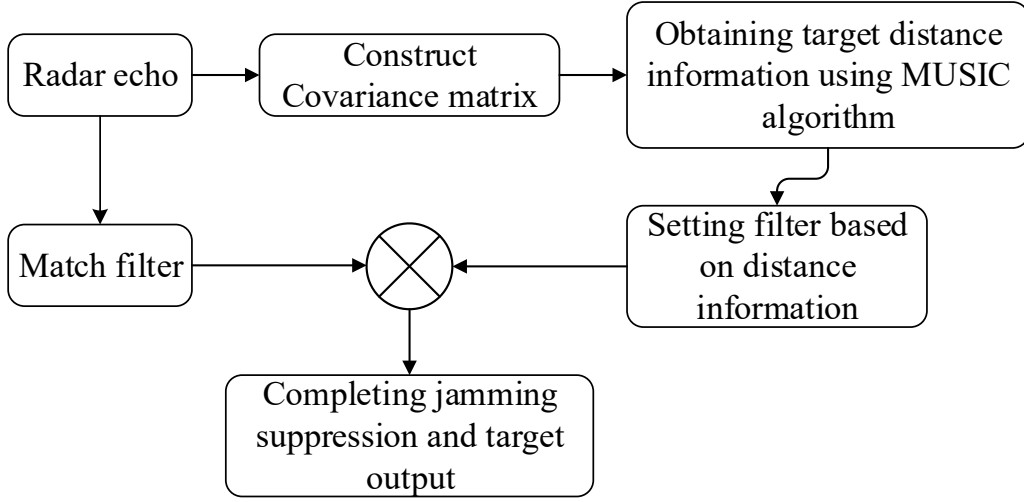

**Figure 9.** Main lobe smart noise jamming suppression process.

## 5. Algorithm Analysis

### 5.1. Algorithm Effectiveness Analysis

Assume that an LFM signal is emitted by the radar under the following simulation conditions: an initial carrier frequency of 1 GHz, a step frequency of 1 kHz, a pulse duration of 300 us, a pulse repetition period of 600 μs, a pulse number of 16, a bandwidth of 5 MHz, and a sampling frequency of 10 MHz. Assume that the initial distance of the target is 30 km and the signal-to-noise radio (SNR) is −10 dB. Assume that the self-defense jammer intercepts the radar signal with the same time width and bandwidth of the signal for product modulation, convolution modulation, and then a full pulse forward, with a JSR of 80 dB.

Figure 10 shows the scanning results of the MUSIC algorithm at a spatial distance of 0 km to 90 km. The MUSIC spectra of noise convolution jamming and noise product jamming are consistent, and their spatial distances are both indicated to be 30 km, verifying the effectiveness of the spectral analysis super-resolution spatial distance extraction method.

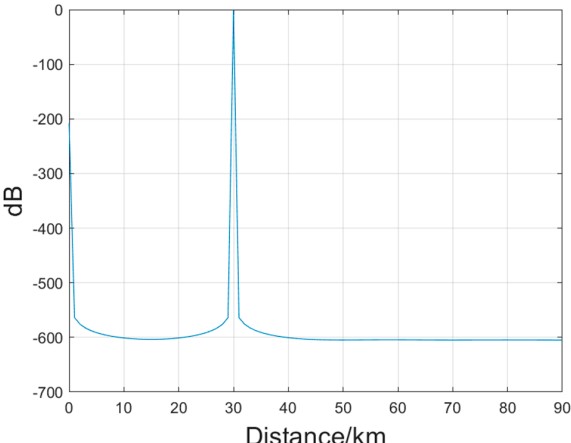

**Figure 10.** Echo spatial distance of the MUSIC spectrum.

Figures 11 and 12 show the two-dimensional distribution of the spatial distance–time domain distance after the pulse pressure of the noise convolution jamming and the noise product jamming, respectively. It can be seen that both types of jamming cover the target signal in the time domain, resulting in a strong suppression effect. This is consistent with conclusion 1 of the previous analysis of smart noise jamming. At the same time, the target and the jamming have the same spatial distance because the internal time–frequency and phase relations of the two are consistent, which is consistent with conclusion 2 of the analysis. The simulation results verify the rationality of this paper's analysis of smart noise characteristics.

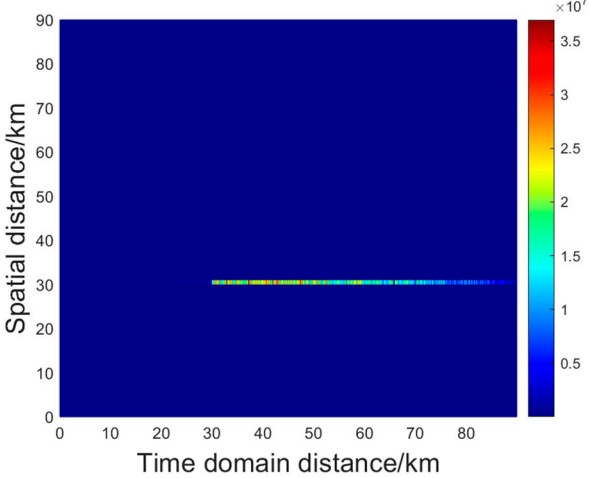

**Figure 11.** Two-dimensional distance distribution spectrum of noise convolution jamming.

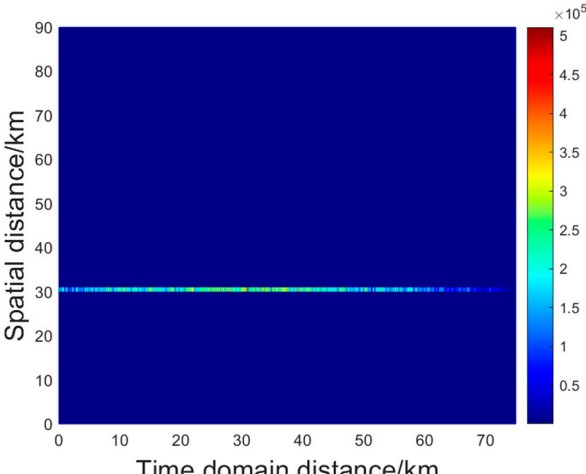

**Figure 12.** Two-dimensional distance distribution spectrum of noise product jamming.

After obtaining the spatial distance information, a time-domain distance filter–Singer function filter can be constructed as follows:

$$h(R) = \sin c(R - R_0) = \frac{\sin[\pi(R - R_0)]}{\pi(R - R_0)} \tag{25}$$

Figure 13 shows the suppression effect of the filter on noise convolution jamming. As can be seen from the figure, the signal-to-jamming noise ratio (SJNR) of the output after jamming suppression has improved by nearly 30 dB, while the signal is actually not lost because the JSR of the jamming is 80 dB, and even the energy of the suppressed jamming is still much higher than the target. At this time, a large part of the output signal is jamming wothin the same distance unit as the target signal.

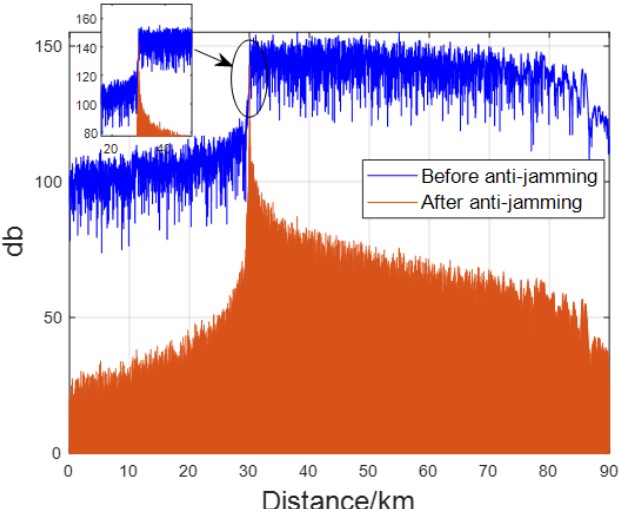

**Figure 13.** Comparison of before and after noise convolution jamming suppression.

Figure 14 shows the suppression effect of the filter on noise product jamming. The suppression effect on noise product jamming is similar to the effect on noise convolution jamming.

The simulation results verify the effectiveness of this method in suppressing smart noise jamming. For noise convolution jamming and noise product jamming, this method can still effectively suppress JSRs of up to 80 dB.

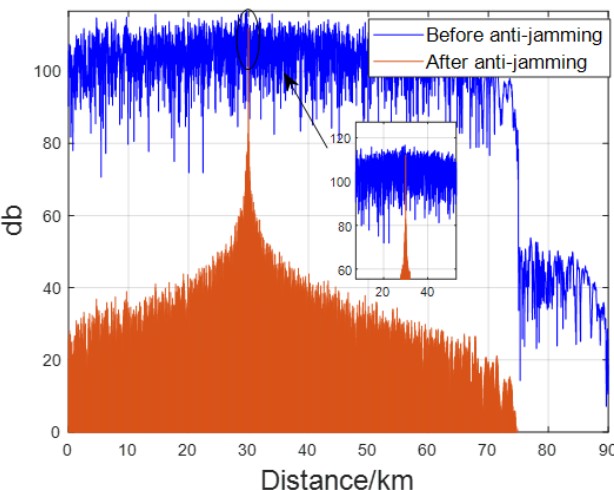

**Figure 14.** Comparison of before and after noise product jamming suppression.

### 5.2. Analysis of Key Factors in Algorithms

The key to this algorithm is how to accurately obtain the spatial distance of the target. As long as the spatial distance of the target can be accurately obtained, a filter can be constructed to suppress jamming. The method of obtaining the target spatial distance in this paper is to use the MUSIC algorithm, the accuracy of which is related to the SNR of the signal and the number of snapshots.

Due to the fact that the spatial distance of smart noise jamming is consistent with the target, the jamming is equivalent to the target in the process of spectral analysis, and the jamming energy is much stronger than the signal energy. Therefore, what truly affects the accuracy of this algorithm is the JSR. Therefore, the fixed SNR is $-10$ dB, and the number of snapshots is 300. If the JSR is set from 0 dB to 80 dB, 50 Monte Carlo simulations may be performed to obtain the average value of the spatial distance error.

As shown in Figure 15, when the SNR is $-10$ dB, the algorithm can stably perform its performance when the JSR is higher than 20 dB, fully meeting the practical application needs.

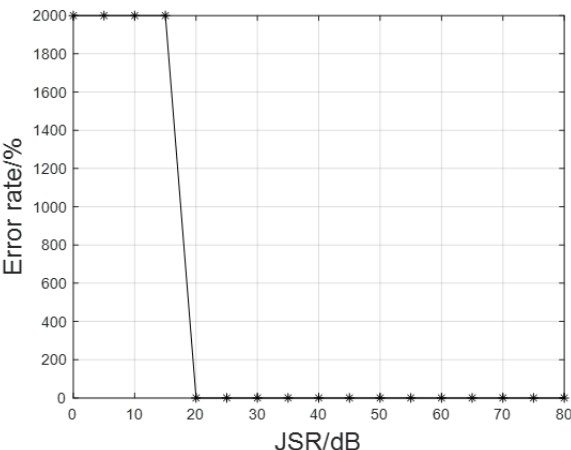

**Figure 15.** The impact of JSR on spatial distance estimation.

The fixed JSR is 30 dB. If the impact of the number of snapshots on the spatial distance estimation is studied, the number of snapshots may be set from 10 to 200, and 50 Monte Carlo simulations may be performed to obtain the average value of the spatial distance error.

As shown in Figure 16, when the JSR is 30 dB and the number of snapshots is only 30, the algorithm can still accurately estimate the spatial distance of the target, demonstrating its good robustness.

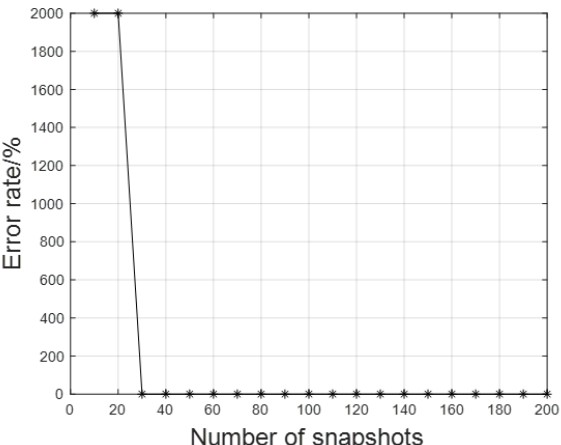

**Figure 16.** The impact of the number of snapshots on spatial distance estimation.

## 6. Conclusions

In this paper, the time–frequency characteristics of smart noise jamming methods, such as noise convolution jamming and noise product jamming, and their correlation with target signals are studied. The essential characteristics of these jamming methods are summarized: they are a series of continuous and random amplitude impulse functions that modulate radar signals in the frequency domain or time domain, respectively. At the same time, this paper proposes a new method of robust self-defensive smart noise jamming suppression based on pulse frequency stepping. The advantage of this method is that it can deal with smart noise jamming with a high JSR, and it is effective for both noise convolution jamming and noise product jamming. The core of this method is to use the spatial distance information of smart noise jamming to be consistent with the radar signal and to obtain the spatial distance information of the target by using the phase difference between adjacent pulses of frequency-stepping pulse trains in order to construct a filter to suppress jamming in the radar echo. The simulation results verify the effectiveness of this method. It is worth mentioning that the method cleverly utilizes the consistent characteristics of the jamming and the target in spatial range spectrum, using jamming to estimate the distance information of the target. Therefore, it has good robustness and can still effectively exert detection efficiency in the face of strong jamming.

**Author Contributions:** Conceptualization, Y.W. and Z.Z.; methodology, Y.Z. and Z.Z.; validation, Z.Z. and B.Z.; resources, B.Z. and H.C.; data curation, Y.Z. and H.C.; writing—original draft preparation, Y.Z.; writing—review and editing, Y.Z. and B.L.; supervision, Z.Z and B.L. All authors have read and agreed to the published version of the manuscript.

**Funding:** This research was funded by the National Nature Science Foundation of China, grant number 62101593 and 62001510.

**Data Availability Statement:** Not applicable.

**Conflicts of Interest:** The authors declare no conflict of interest.

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
