# Peer review of "Analysis of Characteristics and Suppression Methods for Self-Defense Smart Noise Jamming"

_electronics, doi:10.3390/electronics12153270_

Round 1
Reviewer 1 Report
The authors are analyzing the correlation between smart noise jamming methods, in particular noise product jamming and noise convolution jamming, and target signals in radars and propose a self-defense suppression technique based on pulse frequency stepping. The paper is decently well-written and well-organized. The topic could be of interest to the readers of MDPI Electronics.
My major concern is that the paper lacks true experimental data. The authors claim to provide experimental evidence in terms of simulation, but in my opinion this is not sufficient and some real hardware measurements would more firmly back the authors' claims and provide additional evidence and validate the proposed technique.
Furthermore, the authors do not even explicitly state how and in which environment (MATLAB or some other program) the simulations have been performed.
I reckon that providing real experimental characterization is mandatory at this level. Supplementing simulation results with detailed description of the environment and how the readers can repeat them is also highly beneficial.
Finally, the authors use MUSIC algorithm to obtain the spatial distance of the target. I am wondering if some other (not so advanced) algorithm is used, how the obtained results would compare to the presented ones. Can you please analyze at least one or two additional algorithms besides MUSIC and comment the obtained results.
On the minor side, please correct the following:
Red line in Fig. 1(a) is barely distinguishable. Please use thick(er) line to represent the intercepted radar signal on both Figures 1 and 2.
Top left block box in Fig. 9 should contain "Radar echo" not "Eadar echo".
Please proof read your paper for grammar. There are some mistakes such as in line 26 of the manuscript:
"...the jammer can rapidly intercepted radar signals, modulated and forwarded them."
whereas it should be:
"...the jammer can rapidly intercept radar signals, modulate and forward them."
then at line 327:
"the output signal is an jamming in the same distance"
where there's no need for "an" because the next word does not begin with a vowel.
Author Response
Response to Reviewer 1 Comments
Point 1: My major concern is that the paper lacks true experimental data. The authors claim to provide experimental evidence in terms of simulation, but in my opinion this is not sufficient and some real hardware measurements would more firmly back the authors' claims and provide additional evidence and validate the proposed technique.
Response 1: Your opinion is very rigorous, but the confidentiality level of the true experimental data of jammings is relatively high, so it is not suitable to be used in this paper. And this method does not require any prior information to be predicted, and can be applied to conventional stepped-frequency radar. Based on the current development of radar frequency agility technology, this method is available in practical application.
Point 2: Furthermore, the authors do not even explicitly state how and in which environment (MATLAB or some other program) the simulations have been performed.
Response 2: Based on your suggestion, an explanation has been added in the text, and the simulation is conducted in MATLAB.
Point 3: I reckon that providing real experimental characterization is mandatory at this level. Supplementing simulation results with detailed description of the environment and how the readers can repeat them is also highly beneficial.
Response 3: The true experimental data cannot be provided in this article due to confidentiality requirements, but the jamming’s parameters and generation method for simulation is theoretically formulated in the article, the reader can repeat them beneficially.
Point 4: The authors use MUSIC algorithm to obtain the spatial distance of the target. I am wondering if some other (not so advanced) algorithm is used, how the obtained results would compare to the presented ones. Can you please analyze at least one or two additional algorithms besides MUSIC and comment the obtained results.
Response 4: The method of obtaining the target’s spatial distance is FFT method, but FFT’s accuracy depends on the number of pulses, the pulses emitted during actual radar operation are limited, with a common one being 16 pulses. In this case, the accuracy of the estimated spatial distance is poor. For the super-resolution algorithm in spectral estimation, the MUSIC algorithm and Capon's MVM algorithm have similar effects.
Point 5:Red line in Fig. 1(a) is barely distinguishable. Please use thick(er) line to represent the intercepted radar signal on both Figures 1 and 2.
Response 5: Your opinion is very rigorous and has been revised in the manuscript.
Point 6: Top left block box in Fig. 9 should contain "Radar echo" not "Eadar echo".
Response 6: Your opinion is very rigorous and has been revised in the manuscript.
Point 7: There are some mistakes such as in line 26 of the manuscript:
"...the jammer can rapidly intercepted radar signals, modulated and forwarded them."
whereas it should be:
"...the jammer can rapidly intercept radar signals, modulate and forward them."
Response 7: Your opinion is very rigorous and has been revised in the manuscript.
Point 8: Then at line 327:
"the output signal is an jamming in the same distance"
where there's no need for "an" because the next word does not begin with a vowel.
Response 8: Your opinion is very rigorous and has been revised in the manuscript.
Thank you for your valuable comments!

Reviewer 2 Report
The authors presented the time-frequency characteristics of noise convolution jamming and noise product jamming, and their correlation with target signals. This reviewer has the following concerns:
1. What CAD tool was used to perform the time-frequency analysis of the jamming techniques. Please briefly talk about the tool in sec. 3.
2. During the multi-frequency simulation could you please specify the frequencies and also how they are significantly different from each other in terms of frequencies for the both jamming techniques?
3. How spatial variations would affect the multi-frequency distribution of the both jamming techniques?
4. Please, correct the text in Fig. 9 from "Eadar" to "Radar".
N/A
Author Response
Response to Reviewer 2 Comments
Point 1: What CAD tool was used to perform the time-frequency analysis of the jamming techniques. Please briefly talk about the tool in sec. 3.
Response 1: Your opinion is very rigorous. In the sec.3, this paper gives a brief description of the Time–frequency analysis tool STFT and why to choose STFT. (Line 154-166)
Point 2: During the multi-frequency simulation could you please specify the frequencies and also how they are significantly different from each other in terms of frequencies for the both jamming techniques?
Response 2: For the convenience of observing, the multi-frequency simulation uses three carrier frequency: 0, -2 and 2 MHz. Comparing to the frequency difference of 1 kHz between pulses during anti-interference in the following text, a frequency difference of 2 MHz ensures that the time-frequency map is not aliased.
Point 3: How spatial variations would affect the multi-frequency distribution of the both jamming techniques?
Response 3: Strictly speaking, spatial variations won’t affect the multi-frequency distribution of the both jamming techniques, because space and time-frequency are different dimensions. This paper is the situation where the interference and target are at the same spatial angle, and the effectiveness of the anti-interference method in space dimension is the worst. The situation considered in this paper is that within the duration of a set of pulse trains, both the target and interference are at the same spatial angle, which is in line with reality.
Point 4: Please, correct the text in Fig. 9 from "Eadar" to "Radar".
Response 4: Your opinion is very rigorous and has been revised in the manuscript.
Thank you for your valuable comments!

Round 2
Reviewer 1 Report
The authors have partially addressed my concerns.
Reviewer 2 Report
No comments.
N/A